# The Impact of Stakeholders’ Roles within the Livestock Industry on Their Attitudes to Livestock Welfare in Southeast and East Asia

**DOI:** 10.3390/ani7020006

**Published:** 2017-01-25

**Authors:** Michelle Sinclair, Sarah Zito, Clive J. C. Phillips

**Affiliations:** Centre for Animal Welfare and Ethics, School of Veterinary Sciences, The University of Queensland, Gatton, Queensland 4343, Australia; m.sinclair6@uq.edu.au (M.S.); s.zito@uq.edu.au (S.Z.)

**Keywords:** animal welfare, attitudes, slaughter, transportation, livestock stakeholders, Asia

## Abstract

**Simple Summary:**

Improving stakeholder attitudes to livestock welfare may help to facilitate the better welfare that is increasingly demanded by the public for livestock. Knowledge of the existing attitudes towards the welfare of livestock during transport and slaughter provides a starting point that may help to target efforts. We compared the attitudes of different stakeholders within the livestock industries in east (E) and southeast (SE) Asia. Farmers were more motivated to improve animal welfare during transport and slaughter by peer pressure, business owners by monetary gain, and business managers by what is prescribed by their company. Veterinarians showed the most support for improving animal welfare. The results suggest that the role that stakeholders play in their sector of the livestock industry must be considered when attempting to change attitudes towards animal welfare during transport and slaughter.

**Abstract:**

Stakeholders in the livestock industry are in a position to make critical choices that directly impact on animal welfare during slaughter and transport. Understanding the attitudes of stakeholders in livestock-importing countries, including factors that motivate the stakeholders to improve animal welfare, can lead to improved trade relations with exporting developed countries and improved animal welfare initiatives in the importing countries. Improving stakeholder attitudes to livestock welfare may help to facilitate the better welfare that is increasingly demanded by the public for livestock. Knowledge of the existing attitudes towards the welfare of livestock during transport and slaughter provides a starting point that may help to target efforts. This study aimed to investigate the animal welfare attitudes of livestock stakeholders (farmers, team leaders, veterinarians, business owners, business managers, and those working directly with animals) in selected countries in E and SE Asia (China, Thailand, Viet Nam, and Malaysia). The factors that motivated them to improve animal welfare (in particular their religion, knowledge levels, monetary gain, the availability of tools and resources, more pressing community issues, and the approval of their supervisor and peers) were assessed for their relationships to stakeholder role and ranked according to their importance. Stakeholder roles influenced attitudes to animal welfare during livestock transport and slaughter. Farmers were more motivated by their peers compared to other stakeholders. Business owners reported higher levels of motivation from monetary gain, while business managers were mainly motivated by what was prescribed by the company for which they worked. Veterinarians reported the highest levels of perceived approval for improving animal welfare, and all stakeholder groups were least likely to be encouraged to change by a ‘western’ international organization. This study demonstrates the differences in attitudes of the major livestock stakeholders towards their animals’ welfare during transport and slaughter, which advocacy organisations can use to tailor strategies more effectively to improve animal welfare. The results suggest that animal welfare initiatives are more likely to engage their target audience when tailored to specific stakeholder groups.

## 1. Introduction

Agribusiness is a large and important global industry that impacts on the lives of over 25 billion animals annually (excluding fish and invertebrates) [1], far larger than that of any other industry. Slaughter and transport are key events for the welfare of the animals involved. As well as impacting the animals, adverse welfare events occurring during slaughter and transport activate adrenergic mechanisms within the body, resulting in increased muscle glycogenolysis and reduced carcass quality [2]. Slaughtermen, livestock transporters, business owners, business managers, farmers, and vets are required to make decisions within their roles that have the ability to improve or jeopardize the welfare of the animals in their care. According to the Theory of Planned Behavior, understanding attitudes is the precursor to understanding human behavior [3]. Understanding the factors that motivate human behavior is of critical importance when trying to encourage behavioral changes that will improve animal welfare. The benefit of understanding the target audience is well understood in terms of improved engagement with a product in marketing spheres [4], but the same understanding seems to rarely have been prioritized when encouraging engagement with an idea, message, or practice in social progress initiatives. 

An understanding of the attitudes of and factors motivating specific groups of stakeholders within an industry could potentially provide advice on how to tailor initiatives for the individuals in each stakeholder group to best encourage improvement in animal welfare.

The effects of nation and culture on attitudes towards animal welfare and the discovery of national differences in motivating factors have been previously reported [5]. That study suggested that progress initiatives in not-for-profit advocacy groups would benefit from being designed for individual nations, using a knowledge of specific attitudes and motivating factors in different regions and cultures. Other research has also yielded evidence of geo-political influences on attitudes to animal welfare related topics. For example, significant consistencies were found in attitudes to animals in university students across 12 Eurasian countries, based on the geo-political region of the students, rather than on other demographic factors such as religion, ethnicity, or age [6]. Similarly, the attitudes of the public in Germany, the USA, and Japan towards animals differ significantly [7]. 

The nature of a person’s involvement with the livestock industry has been associated with differing attitudes to animal welfare. For example, discord in attitudes to animal welfare exists in Belgium between farmers, who reported satisfactory levels of farm animal welfare, and the public, who described the current state of farm animal welfare as ‘problematic’ [8]. This highlights the importance of understanding and improving the attitudes to animal welfare of each key stakeholder group within the industry. Understanding the attitudes and resulting behaviors of the stakeholders directly involved in handling animals is of particular importance given the direct impact that their attitude has on animal welfare [9,10,11]. The consumers’ growing demand for high welfare products in developed countries [8] has implications if products come from international trade with countries that have lower welfare practices [12].

There are no reports of attitudes of stakeholders in different livestock industry roles within Asia. When livestock are sent there from developed countries, such as Australia, there is considerable concern by those internal and external to the industry that advocate, on behalf of the animals, that the animal welfare standards are poor [13]. Asia is responsible for 39% of global livestock production [1], and the industry is growing rapidly on that continent. In addition to supplying the domestic market, Asian producers are now also exporting animal products, such as Thai chicken exports to the European Union. Thus the export markets include nations with legislated welfare requirements and consumers who demand products with high standards of animal welfare. This is a critical time for the development of the livestock industry in many Asian countries, yet little is known about the attitudes towards animal welfare of the people who work within the industry. 

While the attitudes of the people concerned with livestock (e.g., veterinary and animal science students [14]) towards animal welfare have been compared between Asian countries, there have been no comprehensive studies of the attitudes of different industry stakeholders towards livestock welfare during transport and slaughter. The aim of this study was to assess both their attitudes to animal welfare and the factors that motivate or act as barriers to improving animal welfare. This information could potentially facilitate the tailoring of animal welfare initiatives to specific stakeholder groups in order to improve stakeholder engagement and address the key welfare issues.

## 2. Methods

Trainers (*n* = 44) with relevant livestock industry knowledge in four key E and SE Asian countries attended one of four two-day workshops (one in each country) presented by four international experts in livestock transport and slaughter. Each workshop included a presentation and explanation of the translated educational resources that were prepared for this project (www.animalwelfarestandards.org). The trainers then delivered forty-four one-day regional workshops to stakeholders (about 25 participants in each) in the livestock transport and slaughter industry in geographically-relevant locations of People’s Republic of China (hereafter China, trainer *n* = 16), Malaysia (*n* = 6), Thailand (*n* = 11), and the Socialist Republic of Viet Nam (hereafter Viet Nam, *n* = 11). These countries were selected because of their important role in global livestock import and export industries and in order to investigate attitudes across countries with diverse religious and cultural attributes. Stakeholders were invited to the workshops and also to participate in the research by the workshop trainers, with the only selection criteria being that they must be employed and involved in the local livestock slaughter and transport industry. The selection criteria to attend the training and participate in the research were given to the trainers, who then invited local stakeholders, most commonly by approaching local businesses and local contacts. 

The invited participants, comprised of individuals involved in animal production industries in their country of residence included slaughtermen, transporters, livestock slaughter and transport business owners and managers, senior livestock veterinarians, and government veterinary representatives.

They were surveyed using a paper-based questionnaire at the start of the slaughter and transport workshops, which had been developed in English through consultation with academic and industry experts in the animal welfare field and through extensive literature review. It was translated into Bahasa, Mandarin, Thai, and Vietnamese and then back translated to ensure consistency of meaning. This questionnaire was also administered to the trainers at the start of their workshops, so that they were familiar with it and because they were also deemed to effectively be stakeholders in the industry. These were incorporated with the stakeholders’ responses, increasing the total number of respondents to 1066.

In the questionnaire, respondent demographics were first obtained, including sex, age, residential area, religion, their self-identified role within the industry, and how their industry knowledge was gained (formal qualifications or otherwise). The options for identified role (and the number of representatives) were: working directly with animals (*n* = 345); team leaders, supervising people who work directly with animals (*n* = 147); business owners (*n* = 55); business managers (*n* = 91); farmers (*n* = 179); veterinarians, who were directly involved in treating animals (*n* = 107); and veterinarians working for the government as an advisor (*n* = 138). 

The rest of the questionnaire consisted of four key question sets with responses to each question being measured on a Likert scale from ‘strongly disagree’ to ‘strongly agree’. 

The first non-demographic set of questions focused on general attitudes to animal welfare, including:
the importance placed on animal welfare during slaughter and transporthow satisfactory animal welfare in the respondents’ workplace was believed to bewhether the respondent intended or felt confident to make animal welfare improvements in their workplacewhether the respondent had tried to make animal welfare improvements in the past

The second question set investigated the key factors influencing the stakeholders’ evaluation of animal welfare during slaughter and transport. These included religion, personal beliefs, the extent to which there are more pressing issues in the community, personal and community monetary gain, importance within the workplace and amongst peers, knowledge, and the relevant laws.

The third question set focused on the respondents’ evaluation of their ability to improve animal welfare during slaughter and transport and the factors that may enable or hinder their ability to effect improvement. These included the same factors as the second question set, but with the addition of company approval of improving animal welfare, physical workspace, available tools and resources, and vehicle design (for transport only).

The final question set focused on sources of encouragement to improve animal welfare and which sources respondents were more likely to respond to favorably. Those investigated were:
prescription by local government, local organizations, local law enforcement, and ‘western’ international organizationsprescription by law, workplace, supervisor, and community eldersthe respondent seeing moral or monetary gain in change or seeing others making the change

The survey was reviewed by a panel of sociological researchers, piloted with nationals from each participating country and amended to ensure comprehension and relevance. 

Once at the workshop, stakeholders were asked by their trainer if they would participate in the research; the only selection criteria were that they must be employed and involved in the local livestock slaughter and transport industry. 

## 3. Statistical Analysis

Multivariable logistic regression analyses were performed using the statistical package Minitab to assess the significance of the relationships between the respondent demographics (with nation, stakeholder role, age, sex, religion, residential zone, and knowledge acquisition method as the independent variables) and the distribution of the Likert scale responses for each question (the dependent variable). Religiosity and length of time in the industry were excluded from the model as their inclusion prevented it from converging after 20 iterations. Models of attitudinal statements utilized all seven stakeholder groups identified, but, when evaluating influences on attitudes, the stakeholder groups were reduced to four logical groups to improve the effectiveness of the logistic regression modelling process; business owners/managers (*n* = 127), private practice/government veterinarians (*n* = 207), the team leader and the staff working directly with animals (*n* = 423), and farmers (*n* = 153). The reference category in both cases was chosen as the most numerous response category (i.e., those working directly with animals in the case of seven groups and the team leader and those working directly with animals in the case of the four groups).

The least squared means of rated importance for each question were determined for each stakeholder group. The importance rankings of factors influencing attitudes to animal welfare were determined using the Fisher Least Significant Difference (LSD) Method, and 95% confidence intervals were determined using the original seven categories of stakeholders. The residuals were checked to ensure they approximated normality. The probability values were considered significant at *p* < 0.05.

This paper focuses on the influence of the stakeholders’ role within the livestock industry on attitudes to animal welfare and the factors that motivate or act as barriers to improving animal welfare. It compares the responses of stakeholders working directly with the animals, supervising other staff members within the industry, business owners or managers within the industry, and livestock veterinarians and farmers. These stakeholders were from China, Malaysia, Thailand, and Vietnam, with the differences between the four nations presented separately [5]. 

## 4. Results

All of the stakeholders participating in the workshops completed the questionnaire at the start of the 44 workshops, yielding 1022 respondents (100% response rate). Three surveys with incomplete data were excluded from analysis, therefore 1019 responses were analysed from the workshop attendees, plus those of the 44 trainers (total *n* = 1063). 

The majority of respondents (*n* = 684; 69%) were male and aged between 26 and 35 (*n* = 361; 36%) or 36–45 (*n* = 248; 25%), with 16% (*n* = 166) being under 25, 15% (*n* = 150) between 46–55, and 6% (*n* = 63) over 56. The majority of the respondents (*n* = 563; 60%) reported that they gained their knowledge through formal qualifications in agriculture and 37% (*n* = 354) through farm employment. Most respondents resided in rural (*n* = 421; 42.5%) or urban (*n* = 398; 40%) areas, with just 17% (*n* = 168) residing in a metropolitan area. Of the 991 respondents who identified their theological affiliation, 43% (*n* = 431) identified as Buddhist, 37% (*n* = 370) as atheist, 7% (*n* = 76) as Muslim, and 4% (*n* = 43) as Christian. 

### 4.1. Attitudes to Animal Welfare during Slaughter and Transport

Respondents who were team leaders and business owners agreed more that the welfare of the animals during slaughter was important to them compared to the respondents working directly with the animals (Table 1). Compared to the respondents working directly with the animals during slaughter and transport, the farmers disagreed more that the welfare of the animals during slaughter and transport was important to them. 

Compared to respondents working directly with the animals during slaughter, veterinarians who practised in the livestock industry agreed more that the welfare of the animals during transport was important to them and that most people who were important to them would approve of them making improvements to the welfare of the animals in their care.

Respondents who were business owners and business managers agreed more that the welfare of animals during slaughter and transport was satisfactory in their workplace, compared to respondents working directly with the animals during slaughter and transport. Business owners agreed more that they had tried to make improvements to the welfare of the animals in their care, compared to respondents working with the animals during slaughter and transport. 

### 4.2. Influencing Factors

Compared to respondents working directly with the animals, farmers agreed less that ‘the importance of the welfare of animals to the company they work for’ was a key factor influencing their personal evaluation of animal welfare or a factor influencing their perceived ability to improve animal welfare during slaughter and transport (Table 2). Conversely, farmers agreed more than those working directly with the animals during slaughter and transport that the importance of animal welfare to those who worked with them influences their personal evaluation of the welfare of animals during slaughter. Farmers also agreed more that the importance of animal welfare to those they work with was one of the main factors influencing their ability to make improvements to animal welfare during transport, as compared to respondents working directly with the animals or team leaders. Farmers agreed less that personal monetary gain, their knowledge of animal welfare, their workspace, and the availability of tools and resources were influencing factors when considering their ability to improve animal welfare during slaughter, as compared with respondents working directly with the animals. 

Compared to respondents working directly with the animals or team leaders, business owners and managers agreed less that their work space was one of the main factors influencing their ability to make improvements to animal welfare during transport.

### 4.3. Sources of Influence to Improve Animal Welfare

Respondents in all roles strongly agreed that they were more likely to change their practices if they were prescribed by law and if they saw moral value in changing them and least likely if prescribed by an international organization (Table 3). Business owners and farmers were as likely to change if they saw the potential for monetary gain as they were for reasons of moral value and the law, whereas monetary gain was of little influence for the other groups. Business managers, farmers, and government veterinarians ranked the importance of company prescriptions as high as moral value and the law, whereas this was a secondary influence for those working directly with animals, team leaders, business owners, and private practice veterinarians. 

## 5. Discussion

This study has demonstrated significant relationships between attitudes to animal welfare and the roles of stakeholders within the livestock industry. The results have implications for the development of initiatives focused on improving animal welfare in these regions; initiatives may be more engaging and successful if they are targeted to specific stakeholder groups, as well as tailored by country [5]. We have also demonstrated that legislation and ‘seeing moral value’ in implementing animal welfare change is shared by several roles; however some important differences exist. Many of these key role differences are in contrast to those working directly with the animals during slaughter and transport.

### 5.1. Business Owners, Team Leaders and Managers

Team leaders and business owners attributed higher levels of importance to the welfare of animals in transport and slaughter as compared to stakeholders in other roles. This high level of importance placed on animal welfare may represent a greater level of understanding and awareness of the concept of animal welfare, its importance to buyers and consumers, and its implications for income. This occurs as the more senior stakeholders, leaders, and owners are tasked with making decisions for the business, the animals, and other stakeholders working with the animals. The apparent importance placed on animal welfare by this group represents a promising opportunity to influence animal welfare improvements in business from the top down. 

Business owners and managers also had a more positive opinion of animal welfare in their workplace, compared to respondents working directly with animals, as they more commonly reported that animal welfare was satisfactory during both slaughter and transport in their workplace. It is possible that business owners and managers feel compelled to report high levels of regard for animal welfare, given their senior roles within industry and the need to create and maintain a positive image for their business. Business owners have greater stakes in the viability and success of their business (and therefore business image) than those working in less senior roles, which may explain the high importance placed on monetary gain as a factor influencing business owners to improve animal welfare. It is of note that business managers were more likely to change their practices to improve animal welfare under the influence of company prescriptions, as compared with owners. This may be explained by the common structure of business operations that often sees business owners as responsible for setting company prescriptions and holds business managers primarily accountable for ensuring that what is prescribed is implemented within the operation.

Business owners and managers were also more likely to report that they had tried to improve animal welfare in the past than those working hands on with the animals. These stakeholders may feel significantly more empowered to engender change, but it is possible that this response was biased by a desire to embody the most desirable business image. Business managers did not see their workspace as an issue when considering their ability to improve animal welfare, whereas people working directly with animals did. This indicates that business managers believe they have or have provided all the necessary tools to improve animal welfare in their business, and this belief is consistent with their positive opinion of animal welfare in their workplace. 

These findings suggest that building awareness amongst business managers, owners, and supervisors about the benefits of improving animal welfare may increase the likelihood of engaging business owners and managers in encouraging employees in efforts to improve animal welfare. Business owners could be encouraged to incorporate higher welfare standards into key performance indicators and initiate company-based training workshops to facilitate and empower improvements to animal welfare during slaughter and transport in their business. Since company prescription was an important motivator for business managers to improve animal welfare, once business owners set higher welfare standards for the company, business managers are likely to engage with implementing these changes. 

### 5.2. Farmers

Farmers placed a lower importance on animal welfare during transport and slaughter compared with other stakeholders; this may be because they are less likely to be involved in these processes. This may also explain why farmers reported that monetary gain, company approval, their knowledge levels, and the availability of tools and resources were less likely to influence their ability to make improvements to animal welfare during slaughter. Farmers also reported that company prescriptions did not influence their ability to improve animal welfare but reported a significantly greater influence of peer acceptance on their evaluation of animal welfare and their ability to make improvements to animal welfare, compared to those working directly with the animals during slaughter and transport. This finding is consistent with a study of Danish dairy farmers that showed social and peer influences to be the most important motivator to comply with environmental policy, outranked only by civil duty [15]. Similarly, a study of Welsh and English cattle farmers found non-supportive peer practices and norms to be a major barrier to the implementation of zoonotic control programs [16]. This could be explained by farmers’ operational and geographic isolation, in which their neighboring farmers and peers may provide their nearest or only point of reference and also the most personally meaningful. This suggests that farmers may be more readily engaged in initiatives that share welfare practices and knowledge from farmer to farmer. This is supported by the results of another study that found that farmers' involvement in active peer dialogue networks to be improved their capacity to innovate [17].

### 5.3. Veterinarians

Veterinarians working with animals in the field reported that the welfare of the animals was more important than did those working hands-on in slaughter and transport and had more respect for peer approval when looking to improve animal welfare. This supports the notion that animal welfare is seen as an inherent role and responsibility of a veterinarian and the stance of the World Animal Health Organisation (OIE) that the world community of veterinarians is dedicated to improving animal welfare [18]. In addition, veterinarians are focused on animal centric measurements and diagnostics, and may therefore be more acutely aware of and knowledgeable about suffering. 

Given the high importance of perceived peer acceptance for actively attempting to improve animal welfare reported by veterinarians, these stakeholders have great potential to be advocates for the animals and drivers of change within the livestock industry. The ability to effectively undertake this role as a driver of animal welfare improvements may be hampered by a conflict of interest for veterinarians working in a private livestock enterprise, where future income may rely on acquiescence with current practice and not entering into conflict with business owners [19]. Thus monetary gain was attributed greater importance by veterinarians who work directly with livestock within a private enterprise, compared to government veterinarians who attributed more importance to company prescriptions, such as policies and standards. This suggests that veterinarians may be best placed to advocate for improved animal welfare when their income is independent of the livestock business that they are advising or monitoring.

The involvement of a ‘western’ international organization was amongst the least likely of the factors to encourage change for each stakeholder role and was overall the least influential source of encouragement. This is particularly the case for those working directly with livestock during slaughter and transport, but this may be, in large part, due to the decreased likelihood of these stakeholders interacting with an international organization. This potentially results in the perception among these stakeholders that an international organization is unlikely to understand their work and so would not be in a position to influence or interfere with their practices. This suggests that improvements to animal welfare may be best initiated locally and within companies.

The potential limitations of this study include response biases towards perceived preferred answers by the respondent on a sensitive issue. Attempts to mitigate this issue were made through the confidential nature of the survey, a request to be honest, and conducting of the survey before any training content was shared. This was also intended to mitigate any trainer bias in the results. We also acknowledge that there are other factors influencing the attitudes expressed in this survey, in particular each respondent’s country (evaluated in [5]), but expect that the close geographical proximity of the four countries leads to greater similarity than between unconnected ‘Western’ and developing countries. 

## 6. Conclusions

This study demonstrated that the attitudes of industry stakeholders in the Asia Pacific region to animal welfare during slaughter and transport are influenced by each stakeholder’s role within the industry. This is particularly the case in regard to the influence of peers, monetary gain, and the company. Legislation and seeing moral value in changes to animal welfare were key motivators to change practices across all stakeholder roles. Key differences between stakeholder groups need to be understood and incorporated in animal welfare initiatives aimed at each target audience to maximize engagement and success. Livestock stakeholders are in a position to make daily decisions that directly affect the welfare of animals during slaughter and transport. Improved understanding of the stakeholder audience and what motivates them should enable the development of more collaborative and engaging initiatives that are likely to succeed in improving the welfare of animals during the critical moments of slaughter and transport. 

## Figures and Tables

**Table 1 animals-07-00006-t001:** Least square means of the Likert scale responses for statements about animal welfare during transport and slaughter in industry respondents from China (*n* = 381), Thailand (*n* = 307), Malaysia (*n* = 124), and Vietnam (*n* = 210). Results indicate the odds ratio, confidence interval, and probability of stakeholders working directly with animals compared with each of the six other groups, in turn, agreeing with the 6 statements in bold.

	Mean Likert Scale Response Value ^1^	Odds Ratio ^2^	95% Confidence Interval ^2^	*p* Value ^2^
**The welfare of the animals during slaughter is important to me.**				
Working directly with the animals	3.93			
Team Leader: supervising people who work directly with the animals	4.27	0.65	0.43–0.98	0.04
Business owner	4.22	0.53	0.29–0.99	0.04
Business Manager	4.05	0.83	0.51–1.35	0.45
Farmer	3.75	2.07	1.33–3.21	<0.001
Veterinarian who treats animals hands on	4.23	0.65	0.41–1.05	0.07
Veterinarian working for the government as an advisor	4.32	0.87	0.52–1.47	0.59
**The welfare of the animals during transport is important to me.**				
Working directly with the animals	3.95			
Team Leader: supervising people who work directly with the animals	4.27	0.7	0.46–1.07	0.09
Business owner	4.31	0.65	0.35–1.19	0.16
Business Manager	4.15	0.73	0.45–1.20	0.22
Farmer	3.86	2.15	1.38–3.33	0.00
Veterinarian who treats animals hands on	4.24	0.59	0.37–0.95	0.02
Veterinarian working for the government as an advisor	4.23	0.8	0.48–0.06	0.41
**The welfare of the animals while being slaughtered is satisfactory in my workplace.**				
Working directly with the animals	3.23			
Team Leader: supervising people who work directly with the animals	3.60	0.68	0.45–1.01	0.05
Business owner	3.68	0.45	0.25–0.82	<0.001
Business Manager	3.59	0.45	0.28–0.73	<0.001
Farmer	3.26	1.18	0.77–1.80	0.45
Veterinarian who treats animals hands on	3.41	0.83	0.53–1.30	0.42
Veterinarian working for the government as an advisor	3.68	0.75	0.45–1.24	0.26
**The welfare of the animals while being transported is satisfactory in my workplace.**				
Working directly with the animals	3.28			
Team Leader: supervising people who work directly with the animals	3.52	0.85	0.57–1.28	0.44
Business owner	3.62	0.51	0.28–0.92	0.02
Business Manager	3.53	0.56	0.35–0.91	0.01
Farmer	3.47	1.21	0.79–1.85	0.39
Veterinarian who treats animals hands on	3.43	0.84	0.53–1.32	0.44
Veterinarian working for the government as an advisor	3.66	0.85	0.51–1.41	0.52
**Most people who are important to me would approve of me making improvements to the welfare of the animals in my care.**				
Working directly with the animals	3.68			
Team Leader: supervising people who work directly with the animals	3.84	0.98	0.65–1.48	0.91
Business owner	3.83	0.94	0.51–1.71	0.83
Business Manager	3.65	1.04	0.64–1.69	0.86
Farmer	3.73	1.44	0.93–2.22	0.10
Veterinarian who treats animals hands on	3.96	0.59	0.37–0.94	0.02
Veterinarian working for the government as an advisor	3.86	0.98	0.58–1.65	0.94
**In the past I have tried to make improvements to the welfare of the animals in my care.**				
Working directly with the animals	3.70			
Team Leader: supervising people who work directly with the animals	3.91	0.85	0.55–1.31	0.47
Business owner	4.14	0.42	0.22–0.79	<0.001
Business Manager	3.92	0.61	0.36–1.02	0.06
Farmer	3.81	1.15	0.73–1.82	0.54
Veterinarian who treats animals hands on	3.73	0.76	0.47–1.25	0.27
Veterinarian working for the government as an advisor	3.52	0.96	0.56–1.62	0.86

^1^ The Likert scale was measured from 1 (strongly disagree) to 5 (strongly agree); ^2^ Probability that response for that stakeholder differs from that of those working directly with animals; derived from multivariable ordinal logistic regression models including all demographic factors.

**Table 2 animals-07-00006-t002:** Least square means of Likert scale responses for the statements about influencing factors in the personal evaluation of, and the ability to, improve animal welfare during transport and slaughter, by industry respondents from China (*n* = 381), Thailand (*n* = 307), Malaysia (*n* = 124), and Vietnam (*n* = 210). Results indicate the odds ratio, confidence interval, and probability of stakeholders working directly with the animals differing from each of three other composite groups, in the importance of different influences on animal welfare issues in their workplace.

	Mean Likert Scale Response Value ^1^	Odds Ratio ^2^	95% Confidence Interval ^2^	*p* Value ^2^
**The following factors influence my personal evaluation of animal welfare during slaughter and transport**				
*Importance to peers*				
Working directly with the animals (including team leaders)	3.68			
Business owner + business managers	3.76	0.80	0.54–1.19	0.26
Farmer	3.69	1.66	1.08–2.55	0.02
Veterinarian (private and government)	3.66	1.05	0.72–1.53	0.8
*Importance of animal welfare to the company I work for*				
Working directly with the animals (including team leaders)	3.74			
Business owner + business managers	3.75	1.01	0.68–1.49	0.96
Farmer	3.50	1.79	1.18–2.72	0.00
Veterinarian (private and government)	3.74	0.87	0.60–1.27	0.47
**The following factors impact my ability to make improvements to welfare during slaughter**				
*Monetary gain*				
Working directly with the animals (including team leaders)	3.42			
Business owner + business managers	3.40	1.10	0.75–1.60	0.62
Farmer	3.66	1.65	0.43–0.98	0.04
Veterinarian (private and government)	3.20	1.40	0.98–2.01	0.06
*Company approval*				
Working directly with the animals (including team leaders)	3.75			
Business owner + business managers	3.77	0.93	0.63–1.38	0.73
Farmer	3.55	2.17	1.42–3.31	0.00
Veterinarian (private and government)	3.60	1.23	0.85–1.80	0.27
*My personal knowledge*				
Working directly with the animals (including team leaders)	3.79			
Business owner + business managers	3.69	1.30	0.88–1.94	0.19
Farmer	3.58	1.63	1.06–2.52	0.02
Veterinarian (private and government)	3.83	1.10	0.74–1.62	0.64
*Workspace*				
Working directly with the animals (including team leaders)	3.69			
Business owner + business managers	3.66	1.16	0.78–1.73	0.45
Farmer	3.65	1.92	1.25–2.97	0.00
Veterinarian (private and government)	3.64	1.13	0.77–1.66	0.53
*Available tools and resources*				
Working directly with the animals (including team leaders)	3.83			
Business owner + business managers	3.87	0.91	0.61–1.36	0.65
Farmer	3.62	1.98	1.29–3.04	0.00
Veterinarian (private and government)	3.76	0.97	0.66–1.42	0.85
**The following factors impact my ability to make improvements to welfare during transport**				
*Company approval*				
Working directly with the animals (including team leaders)	3.86			
Business owner + business managers	3.75	1.28	0.86–1.91	0.22
Farmer	3.74	1.85	1.20–2.84	0.00
Veterinarian (private and government)	3.78	1.14	0.77–1.67	0.51
*Importance to peers*				
Working directly with the animals (including team leaders)	3.70			
Business owner + business managers	3.59	1.00	0.67–1.48	0.98
Farmer	3.75	1.72	1.12–2.65	0.01
Veterinarian (private and government)	3.76	1.09	0.75–1.60	0.64
*Workspace*				
Working directly with the animals (including team leaders)	3.64			
Business owner + business managers	3.41	1.65	1.11–2.43	0.01
Farmer	3.82	0.92	0.59–1.43	0.70
Veterinarian (private and government)	3.63	0.98	0.67–1.43	0.90

^1^ The Likert scale was measured from 1 (strongly disagree) to 5 (strongly agree); ^2^ Probability that response for that stakeholder differs from that of those working directly with animals; derived from multivariable ordinal logistic regression models including all demographic factors.

**Table 3 animals-07-00006-t003:** Differences between stakeholder groups in the importance rankings of factors influencing attitudes to animal welfare in Chinese (*n* = 381), Thai (*n* = 307), Malaysian (*n* = 124), and Vietnamese (*n* = 210) industry respondents.

Stakeholder	I Am More Encouraged to Change My Practices by…
Working directly with the animals	Law ^a^, Moral value ^a,b^, Local gov ^b,c^, Police ^c^, Workplace ^c,d^, Local org ^c,d,e^, Supervisor ^d,e,f^, Money gain ^e,^^f^, Peers ^f^, Comm leader ^f^, Intl org ^g^
Team Leader	Moral value ^a^, Law ^a^, Workplace ^b^, Local gov ^b^, Police ^b^, Local org ^b,c^, Supervisor ^b,c,d^, Money gain ^b,c,d^, Peers ^b,c,d^, Comm leader ^c,d^, Intl org ^d^
Business owner	Law ^a^, Moral value ^a,b^, Money gain ^a,b,c^, Local gov ^a,b,c^, Workplace ^b,c,d^, Police ^b,c,d^, Local org ^b,c,d^, Peers ^c,d,e^, Supervisor ^d,e^, Intl org ^d,e^, Comm leader ^e^
Business Manager	Moral value ^a^, Law ^a,b^, Workplace ^a,b,c^, Local gov ^a,b,c^, Local org ^a,b,c,d^, Supervisor ^a,b,c,d^, Police ^b,c,d^, Peers ^b,c,d^, Comm leader ^c,d^, Money gain ^c,d^, Intl org ^d^
Farmer	Moral value ^a^, Law ^a,b^, Local org ^a,b^, Money gain ^a,b^, Local gov ^a,b^, Workplace ^a,b,c^, Peers ^a,b,c^, Police ^b,c^, Comm leader ^c^, Intl org ^c^, Supervisor ^c^
Private practice veterinarian	Law ^a^, Moral value ^a^, Local gov ^a,b^, Workplace ^b,c^, Local org ^b,c^, Police ^b,c^, Supervisor ^c,d^, Comm leader ^d^, Peers ^d^, Money gain ^d^, Intl org ^d^
Government veterinarian	Law ^a^, Moral value ^a^, Workplace ^a,b^, Police ^a,b^, Local gov ^b,c^, Local org ^b,c,d^, Supervisor ^c,d,e^, Intl org ^d,e,f^, Peers ^e,f^, Comm leader ^f^, Money gain ^f^

Factors within stakeholder groups with different superscripts are significantly different (*p* < 0.05); Legend: ‘if changes are prescribed by…’; local government (‘local gov’), a local organization (‘local org’), local law enforcement (‘police’), a ‘western’ international organization (‘intl org’), the law (‘law’), my company (‘workplace’), my supervisor (‘supervisor’), or my community elder or community leader (‘comm leader’); ‘I see…’; moral value in changing practices (‘moral value’), personal monetary gain from changing practices (‘money gain’), or others making the change (‘peers’).

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
