# Peer review of "The Impact of Stakeholders’ Roles within the Livestock Industry on Their Attitudes to Livestock Welfare in Southeast and East Asia"

_animals, 2017, doi:10.3390/ani7020006_

Round 1

Reviewer 1 Report

This is a worthwhile study, well carried out and with an impressive sample size. The only important problem with the paper is that it is not clear how statements conveying or implying statistical significant differences are supported by the statistics: see comments on 4/17-20

1/13     In both the Simple Summary and the Abstract, the desirability of improving attitudes to animal welfare, or animal welfare itself, is taken for granted. Either a brief justification for this should be added, or this approach should be introduced in a conditional way (e.g. ‘If it is desired to improve attitudes …’)

1/15     It would be clearer to emphasise ‘stakeholders within the livestock industries,’ rather than ‘in’ – see comment on 2/31

1/23     trade relations between whom?

1/31, 37           This use of ‘major’ does not seem justified by the results: differences between groups in Tables 1 and 2 are fairly minor. Nor does it seem to be based on the Discussion or Conclusions. This word should be justified where appropriate, or deleted

1/38     Here and elsewhere I suggest putting ‘western’ in inverted commas, as it includes Australasian organisations

1/39     ‘to tailor strategies more effectively’ would be preferable grammatically. The same grammatical issue also occurs in several other places.

2/2       Correct to ‘68 billion animals a year (excluding fish and invertebrates)’

2/10     For clarity, add ‘human’ before ‘behavior’

2/24     Correct to ‘12 Eurasian countries’

2/31     The term ‘stakeholders’ is used in a variable way in this paper. It normally includes any group with an interest in the topic, including general public and NGOs. Here the emphasis is on groups WITHIN the industry, so this needs clarifying. Thus as this paragraph has mentioned attitudes of the general public, it would be appropriate to clarify the last sentence by adding ‘within the industry’ after ‘each key stakeholder group.’

2//35    Correct ‘have implications’ to ‘has implications’

2/39     concern by whom?

2/46     Given the comment on 2/31, describing students as ‘stakeholders’ here is again confusing. I suggest rephrasing this as ‘the attitudes of people concerned with livestock’

3/5       Delete ‘workshop’ after ‘countries’

4/15     Correct ‘Religiosity’ to ‘religion’

4/17-20            The statement that ‘The reference category was chosen as the most numerous response category’ appears to relate only to the comparison between four groups described in the previous sentence. However, if I understand Table 1 correctly, that compares eight groups, similarly taking people who worked with animals as a reference category.

            This approach also needs justification. Why are comparisons only made between people who worked with animals and other categories, rather than overall, or pairwise between categories?

            Insofar as that approach is justified, it needs explaining again in the Table captions and perhaps also briefly in Results. As it stands, I found it difficult to understand the support for statements of results such as the first sentence in section 4.1 (5/2), including phrases such as ‘more likely.’

4/31     This reference is number 5 in the list

4/34     Why is this number different from the 1066 in 3/27?

5/35     A reference to Table 3 is needed in the text

Table 3 Use of superscripts to show statistics needs explanation

5/41-5  I do not understand the basis for these two sentences. If I understand correctly, the ranking of factors by government and private vets has not been compared statistically. And in Table 3, the relevant factors only differ in their ranking, between these two categories of vets, by one or at most two places.

7/18     This sentence is unclear

7/30     As in the comment on 5/41, I do not understand the basis of this statement

7/ 35    After ‘likely’ add ‘factors’

Author Response

Reviewer 1

This is a worthwhile study, well carried out and with an impressive sample size.

Authors: Thank you

The only important problem with the paper is that it is not clear how statements conveying or implying statistical significant differences are supported by the statistics: see comments on 4/17-20

Authors: We have addressed that as described below

1/13 In both the Simple Summary and the Abstract, the desirability of improving attitudes to animal welfare, or animal welfare itself, is taken for granted. Either a brief justification for this should be added, or this approach should be introduced in a conditional way (e.g. ‘If it is desired to improve attitudes …’)

Authors: A justification is given in both places: ‘Improving stakeholder attitudes to livestock welfare may help to facilitate the better welfare that is increasingly demanded by the public for livestock.’

1/15 It would be clearer to emphasise ‘stakeholders within the livestock industries,’ rather than ‘in’ – see comment on 2/31

Authors: ‘in’ changes to ‘within’

1/23 trade relations between whom?

Authors: Sentence revised to the following: ‘Understanding attitudes of stakeholders in livestock-importing countries, including factors that motivate the stakeholders to improve animal welfare, can lead to improved trade relations with exporting developed countries, and improved animal welfare initiatives in the importing countries.’

1/31, 37 This use of ‘major’ does not seem justified by the results: differences between groups in Tables 1 and 2 are fairly minor. Nor does it seem to be based on the Discussion or Conclusions. This word should be justified where appropriate, or deleted

Authors: ‘major’ deleted

1/38 Here and elsewhere I suggest putting ‘western’ in inverted commas, as it includes Australasian organisations

Authors: Western placed in inverted commas at the five points of mention in the paper

1/39 ‘to tailor strategies more effectively’ would be preferable grammatically. The same grammatical issue also occurs in several other places.

Authors: changed as requested

2/2 Correct to ‘68 billion animals a year (excluding fish and invertebrates)’

Authors: We have taken the latest FAO data, which suggests that there are over 25 billion farm animals worldwide (with exclusions noted).

2/10 For clarity, add ‘human’ before ‘behavior’

Authors: added

2/24 Correct to ‘12 Eurasian countries’

Authors: corrected

2/31 The term ‘stakeholders’ is used in a variable way in this paper. It normally includes any group with an interest in the topic, including general public and NGOs. Here the emphasis is on groups WITHIN the industry, so this needs clarifying. Thus as this paragraph has mentioned attitudes of the general public, it would be appropriate to clarify the last sentence by adding ‘within the industry’ after ‘each key stakeholder group.’

Authors: added as requested

2//35 Correct ‘have implications’ to ‘has implications’

Authors: corrected

2/39 concern by whom?

Authors: added ‘by those internal and external to the industry that advocate for the animals’

2/46 Given the comment on 2/31, describing students as ‘stakeholders’ here is again confusing. I suggest rephrasing this as ‘the attitudes of people concerned with livestock’

Authors: changed

3/5 Delete ‘workshop’ after ‘countries’

Authors: deleted

4/15 Correct ‘Religiosity’ to ‘religion’

Authors: We measured religiosity (the extent to which a person is religious), as well as actual religion, and the former was not incorporated in the model as its inclusion prevented the model from converging.

4/17-20 The statement that ‘The reference category was chosen as the most numerous response category’ appears to relate only to the comparison between four groups described in the previous sentence. However, if I understand Table 1 correctly, that compares eight groups, similarly taking people who worked with animals as a reference category.

Authors: we have clarified that the number of stakeholder groups was reduced to four just for the logistic regression modelling of influences on attitudes. The entire 7, not 8 groups, were used for the modelling of attitudinal statements.

This approach also needs justification. Why are comparisons only made between people who worked with animals and other categories, rather than overall, or pairwise between categories?

Authors: logistic regression analysis, the usual analytical method for such surveys, always works by comparison to a reference group, which is usually chosen as the most numerous group. It would be possible to make pairwise comparisons by multiple changes in the reference group for each comparison, but it would become very complicated both to undertake and explain.

 Insofar as that approach is justified, it needs explaining again in the Table captions and perhaps also briefly in Results. As it stands, I found it difficult to understand the support for statements of results such as the first sentence in section 4.1 (5/2), including phrases such as ‘more likely.’

Authors: In the table captions, detailed explanations have been added, e.g. Table 1: ‘Results indicate the odds ratio, confidence interval and probability of stakeholders working directly with animals compared with each of the six other groups in turn agreeing with 6 statements in bold’. In the Results we have removed the ‘more likely to agree/disagree or say…’ statements and replaced them with the more direct and accurate comparison: ‘agreed more, or less’. We hope this is easier to understand.

4/31 This reference is number 5 in the list

Authors: the reference has been replaced by number 5.

4/34 Why is this number different from the 1066 in 3/27?

Authors: it did not include the trainers, we have now made that clear, as follows: ‘All of the stakeholders participating in the workshops completed the questionnaire at the start of the 44 workshops, yielding 1022 respondents (100% response rate). Three surveys with incomplete data were excluded from analysis, therefore 1019 responses were analysed from the workshop attendees, plus 44 trainers (total n = 1063).’

5/35 A reference to Table 3 is needed in the text

Authors: this has been added at the start of section 4.3.

Table 3 Use of superscripts to show statistics needs explanation

Authors: We have added: ‘Factors within stakeholder groups with different superscripts are significantly different (P < 0.05)’

5/41-5 I do not understand the basis for these two sentences. If I understand correctly, the ranking of factors by government and private vets has not been compared statistically. And in Table 3, the relevant factors only differ in their ranking, between these two categories of vets, by one or at most two places.

Authors: This is a valid criticism, we did not compare groups in this analysis, only a within group comparison of factors. We have rewritten this section to highlight the major influences on attitudes, which essentially highlighted the biggest and smallest influences initially and then outlined any major divergences from the overall trends in individual groups.

7/18 This sentence is unclear

Authors: we have revised the text to read ‘and had more respect for  peer approval when looking to improve animal welfare.’, which should be more easily understood.

7/30 As in the comment on 5/41, I do not understand the basis of this statement

Authors: We have simplified this sentence, so that it is hopefully now more understandable: ‘Thus monetary gain was attributed greater importance by veterinarians who work directly with livestock within a private enterprise, compared to government veterinarians who attribute more importance to company prescriptions, such as policies and standards.’

7/ 35 After ‘likely’ add ‘factors’

Authors: we have changed to ‘least likely of the factors’

Reviewer 2 Report

in 4.1 of the Results, attitudes to AW during transportation and slaughter, the results of the respondents from governmental representatives should be given since the samplings were taken for them based on the method. 

2)the results presented in table1 and table 2 were the outcomes of the Asian Countries surveyed, I would like to suggest that it may be better to show the result of each individual Country which is much different from each other in culture, religion, education levels etc on the questions given, just like a tourist who visits all these counties would have a big diverse impression on the Country he/she visited. 

Author Response

Reviewer 2

in 4.1 of the Results, attitudes to AW during transportation and slaughter, the results of the respondents from governmental representatives should be given since the samplings were taken for them based on the method.

Authors: these were not included in the options: The options for identified role were (and the number of representatives) were: working directly with animals (n = 345); team leaders, supervising people who work directly with animals (n = 147); business owners (n = 55); business managers (n = 91); farmers (n = 179); veterinarians who were directly involved in treating animals (n = 107); and veterinarians working for government as an advisor (n = 138).

2the results presented in table1 and table 2 were the outcomes of the Asian Countries surveyed, I would like to suggest that it may be better to show the result of each individual Country which is much different from each other in culture, religion, education levels etc on the questions given, just like a tourist who visits all these counties would have a big diverse impression on the Country he/she visited. 

Authors: Results for the different countries have been presented separately, in a paper in Animal Welfare that currently is awaiting minor revision before publication.

Reviewer 3 Report

Main comment: The strong point of this paper is that they include four SE countries and many respondents. A weak point is that they use survey analysis on a sensitive topic such as animal welfare (I wouldn't trust the outcomes as strongly as the authors present them). Another weak point is that they had 44 different workshops with different trainers (even though these were instructed beforehand). Also the selection of participants is not described at all ("the only criteria..." -> is not very descriptive). Still, these survey outcomes may be useful in comparison with other methods that are probably more reliable. So I like it anyway. More detailed comments in chronological order: - The introduction is fine and well written. However, since the entire research depends on the categories you describe, I would appreciate a bit more information about these. E.g. what do you mean by "business owner"? Is a farmer not a business owner, too? And what distinguishes a business manager from an owner, isn't the owner also often the manager? - Method: you used survey data to analyse animal welfare attitudes. Isn't it highly likely that respondents give "polite" answers, answers they think make them look/appear better than they really are? This is a generic issue when using surveys like you did, but for a topic like this I expect it has a huge influence. - The 1st sentence of the method section is weird. Why is it not stated below the acknowledgements or so? (Shouldn't it be part of the meta-data, known to the editor, but not necessarily in the article itself?) - Aren't you afraid that there was a trainer bias because you had a different trainer for every workshop? (Did you analyse that?) - The selection of stakeholders is presented as if they incidentally walked by the researchers, and if they happened to be involved with livestock slaughter or transport, they'd qualify as participant. I can't believe it happened that way. How did you find them / approach them? Are they really representative? - I am very confused by the last paragraph of section 3. I miss the point of the paragraph within this section, and even more since it is the last paragraph of the section. - The results section starts with descriptive statistics. Did you ever use these in connection with any outcomes? - In table one, you didn't highlight the significant outcomes. Perhaps you could? - You probably formulate as precisely and accurately as possible, but to me the sentences used in subsections 4 are tough to read and confusing: (4.1) .... were more likely to disagree that ... was important (4.2) ... were less likely to agree that 'importance' was a key factor (4.3) Business owners' ranked prescription by ...... equally in terms of influence on the likelihood of them changing practices..." [btw, shouldn't the apostroph after "Business owners" go?] etc. Since you take the mean value of Likert scale scores in your analysis (treating the scores as numbers), by the same token you can then turn around "disagree / agree" and "more / less" to make a sentence better readable. - You say that you compared the 4 countries in another publication. So here, you take the countries together and treat them as one. But I wonder whether you can do that just like that. To what extent are the countries comparable, with respect to livestock sector structure? (E.g. supply chain characteristics, amount of slaughters / transporters to choose from, freedom of decision-making / contract obligations, etc etc). Minor comments: - In the very first sentence of the simple summary, I would remove "A" as the first word. A knowledge sounds weird to me. - The indicated corresponding author is specified by the e-mailaddress of another author (why then not give the other author the star?)

Author Response

Reviewer 3

Main comment: The strong point of this paper is that they include four SE countries and many respondents. A weak point is that they use survey analysis on a sensitive topic such as animal welfare (I wouldn't trust the outcomes as strongly as the authors present them). Another weak point is that they had 44 different workshops with different trainers (even though these were instructed beforehand). Also the selection of participants is not described at all ("the only criteria..." -> is not very descriptive). Still, these survey outcomes may be useful in comparison with other methods that are probably more reliable. So I like it anyway. More detailed comments in chronological order: -

Authors: We have now included a section on limitations, which acknowledges some of these weaknesses. The section on participants is much more detailed than before.

The introduction is fine and well written. However, since the entire research depends on the categories you describe, I would appreciate a bit more information about these. E.g. what do you mean by "business owner"? Is a farmer not a business owner, too? And what distinguishes a business manager from an owner, isn't the owner also often the manager? –

Authors: Thank you! The distinction between roles was respondent classified. ..ie – they self-identified with the category. We have now included that in the manuscript for clarity. We have also added to the Method: ‘The selection criteria to attend the training, and therefore participate in the research was given to the trainers, who then invited local stakeholders, most commonly by approaching local businesses and local contacts.’

Method: you used survey data to analyse animal welfare attitudes. Isn't it highly likely that respondents give "polite" answers, answers they think make them look/appear better than they really are? This is a generic issue when using surveys like you did, but for a topic like this I expect it has a huge influence. –

Authors: Agreed, assessing animal welfare attitudes has these challenges. We did attempt to mitigate some of these challenges. I have now included that in the limitations section at the end of the Discussion.

 The 1st sentence of the method section is weird. Why is it not stated below the acknowledgements or so? (Shouldn't it be part of the meta-data, known to the editor, but not necessarily in the article itself?) –

Authors: agreed, we have removed the following: ‘As part of a larger OIE project to improve knowledge around animal welfare standards’, it is already stated in the Acknowledgements section.

 Aren't you afraid that there was a trainer bias because you had a different trainer for every workshop? (Did you analyse that?) –

Authors: This should’ve been mitigated by conducting the survey before any training content was shared, and for this reason we did not include trainer in the model. We have now included this in limitations (in the discussion).

The selection of stakeholders is presented as if they incidentally walked by the researchers, and if they happened to be involved with livestock slaughter or transport, they'd qualify as participant. I can't believe it happened that way. How did you find them / approach them? Are they really representative? –

Authors: we approached a wide range of potential participants (see section 2, paragraph 2) and then gave the respondents a range of options in the questionnaire to self-identify with appropriate groupings of stakeholders, listed as follows: those working directly with animals ; team leaders, supervising people who work directly with animals ; business owners ; business managers ; farmers ; veterinarians who were directly involved in treating animals ; and veterinarians working for government as an advisor. The selection method is now described fully in the Method, together with inclusion of the numbers of respondents in each group.

 I am very confused by the last paragraph of section 3. I miss the point of the paragraph within this section, and even more since it is the last paragraph of the section. –

Authors: we have revised the paragraph to make it clear that this paper is about differences in attitudes arising from the stakeholders’ role, the sister paper is about differences arising from their country.

The results section starts with descriptive statistics. Did you ever use these in connection with any outcomes? –

Authors: we did, but here we report only the effects of stakeholder group. In the statistical analysis section we reported: ‘Multivariable logistic regression analyses were performed in Minitab to assess the significance of the relationships between respondent demographics (nation, stakeholder role, age, sex, religion, residential zone, knowledge acquisition method, as the independent variables) and the distribution of the Likert scale responses for each question (the dependent variable)’

In table one, you didn't highlight the significant outcomes. Perhaps you could? - You probably formulate as precisely and accurately as possible, but to me the sentences used in subsections 4 are tough to read and confusing: (4.1) .... were more likely to disagree that ... was important (4.2) ... were less likely to agree that 'importance' was a key factor (4.3) Business owners' ranked prescription by ...... equally in terms of influence on the likelihood of them changing practices..." [btw, shouldn't the apostroph after "Business owners" go?] etc. Since you take the mean value of Likert scale scores in your analysis (treating the scores as numbers), by the same token you can then turn around "disagree / agree" and "more / less" to make a sentence better readable. –

Authors: In table 1 we have not highlighted the significant results as emboldening these would potentially confuse the reader as the statements are in bold. We have added to the title that the statements are in bold to make it clear.

 We have removed the ‘more likely to agree/disagree or say…’ statements and replaced them with the more direct and accurate comparison: ‘agreed more, or less’. We hope this is easier to understand.

You say that you compared the 4 countries in another publication. So here, you take the countries together and treat them as one. But I wonder whether you can do that just like that. To what extent are the countries comparable, with respect to livestock sector structure? (E.g. supply chain characteristics, amount of slaughters / transporters to choose from, freedom of decision-making / contract obligations, etc etc).

Authors: We acknowledge that the countries are different and have added the limitation to the discussion: ‘We also acknowledge that there are other factors influencing attitudes expressed in this survey, in particular their country (evaluated in [5]), but expect that the close geographical proximity of the four countries leads to greater similarity than between unconnected ‘Western’ and developing countries.’

Minor comments: - In the very first sentence of the simple summary, I would remove "A" as the first word. A knowledge sounds weird to me. –

Authors: ‘A’ has been removed

The indicated corresponding author is specified by the e-mail address of another author (why then not give the other author the star?)

Authors: We have given the other author the star

Round 2

Reviewer 3 Report

I am fine with the modifications the authors made, thank you. So no further comments from me.